# Prevalence and determinants of depression, anxiety, and stress among the elderly population in Bangladesh: A cross-sectional study

Sahabul Haque[1], Suchana Akter[1], Lamia Jannat[1], Zidan Ahmed[1], Mohammad Arifur Rahman[1], Imran Hossain Sumon[1]*, Md. Mahfuzur Rahman[1,2], Md. Salah Uddin[1,3], Md. Moyazzem Hossain[1]*

1 Department of Statistics and Data Science, Jahangirnagar University, Savar, Dhaka, Bangladesh,
2 Department of Management Studies, Jahangirnagar University, Savar, Dhaka, Bangladesh,
3 Department of English, Jahangirnagar University, Savar, Dhaka, Bangladesh

* imran.stat@juniv.edu (IHS); hossainmm@juniv.edu (MMH)

## Abstract

### Background

Mental health issues are commonly faced by the elderly population aged 60 and above, who are influenced by various risk factors, i.e., financial, family, social, and others that impact their quality of life. This study intends to identify the prevalence of depression, anxiety, and stress among the elderly population of Bangladesh and their associated factors.

### Method

This cross-sectional study collected primary data from a sample of 400 elderly individuals aged 60 and above across four districts in Bangladesh. Depression, anxiety, and stress levels were measured using the PHQ-9, GAD-7, and PSS-10 scales, respectively. The ordinal logistic regression model is fitted to evaluate the effects of socio-demographic factors, and a confirmatory factor analysis is used to identify risk factors via a structural equation model.

### Result

Findings revealed that 5.3% of elderly people have severe depression, 9.3% severe anxiety, and 4.75% high perceived stress. Older adults with poor health status were more likely to suffer from severe depression and anxiety. Social isolation and loneliness influence depression in old age. Significant risk factors include poor medical support from family and inadequate family relationships as age progresses, social discrimination and prejudice, and perceived social status. These factors were significantly associated with mental health problems (depression, anxiety, and stress) among the elderly population aged 60 and above.

the Creative Commons Attribution License, which permits unrestricted use, distribution, and reproduction in any medium, provided the original author and source are credited.

**Data availability statement:** All relevant data are within the paper and its Supporting Information files.

**Funding:** The author(s) received no specific funding for this work.

**Competing interests:** The authors have declared that no competing interests exist.

## Conclusion

Depression, anxiety, and stress are significant mental health issues among older people in Bangladesh. To ensure their quality of life, it is essential to diagnose and address these mental health problems to establish effective policies.

## Introduction

The burden of mental problems in older adults is rising due to global population aging. Old age is a life stage that brings various changes to the end of an elderly person's life. Older people face various financial, family, social, cultural, and other challenges as they approach the end of their lives. The main concern for the elderly population in Bangladesh is poor health, and many of them live in poverty their entire lives. Ageing will be a significant issue in Bangladesh due to the country's large population, limited resources, poverty, inadequate medical facilities, and lack of social security [1]. Active ageing is influenced by a range of factors, including marital status, income, decision-making ability, regular walking and physical activity, smokeless tobacco use, newspaper reading as a hobby, medication use, and access to health services [2]. A study also noted that the rapid increase in Bangladesh's elderly population after 2040 will put current healthcare systems and family ties in social security at risk [3]. In Bangladesh, older persons typically encounter greater difficulties with mobility, access to public transportation, health care, and economic support [4]. In Bangladesh, there are no health or social care facilities specifically designed for senior citizens. Rather, the only actions implemented so far are the Old Age Allowance (OAA) for older persons who are financially vulnerable and limited coverage pension plans for government employees [5].

People in old age often encounter various psychological issues such as depression, anxiety, and stress. Depression is a mental health condition in which there is deep discouragement, hopelessness, and sadness, characterized by a loss of motivation in life [6]. In the elderly, depression and anxiety have a significant impact on mental health and quality of life, leading to increased healthcare costs and decreased life quality [7]. Stress refers to the emotional and psychological response when people perceive an overwhelming situation that exceeds their ability to cope [8]. In older people, loneliness is highly associated with increased depressive symptoms and decreased health status over a loneliness time [9]. Because of the AI revolution taking place in industries all over the world, older employees' mental health is impacted by their fear of losing their jobs, which includes anxiety and unease [10]. Different socio-demographic factors, coping styles, self-esteem, and social support explained the variance of 34% in stress for older men and women [11]. Among older people, the prevalence of depression is influenced by factors such as gender, illness, social isolation, and cognitive and functional impairments [12].

The mental health of older people is significantly influenced by various socio-demographic factors, physical health status, and lifestyles living in rural communities [13]. Among the elderly in the United States, a significant association has been

identified between various forms of abuse and mental health issues, which shows robust support is needed to prevent mistreatment. The national mistreatment study reveals that approximately 1 in 10 older people in the United States experience elder abuse, such as emotional, physical, sexual, and financial abuse [14]. The COVID-19 pandemic has increased mental health issues among older people. A study on the COVID-19 outbreak indicates a significant prevalence of depression, anxiety, and stress among elderly individuals who are home-isolated, stay in social isolation, and economic instability is a highly influential factor [15]. The pandemic situation has also heightened depression, anxiety, and stress levels among older people [16]. Physical activity is identified as a preventive factor against mental health challenges among older women. Elderly active women have lower depression, anxiety, and stress compared with inactive women in physical activity [17]. Increasing social engagement can enhance physical and mental well-being [18]. Further, studies about the effect of depression and stress together among older people will improve treatments and health care for them [19].

The prevalence of depression is high among older people who face physical health issues and social support [20]. Similarly, these factors significantly influenced depression, anxiety, and cognitive disorders among urban elderly people in Odisha, India [21]. Due to a variety of socioeconomic and cultural issues, public mental health in South Asia faces numerous obstacles. Mental health is stigmatized and is not given priority as a health concern in this area. The lack of knowledge and comprehension of mental health concerns is one of the largest problems in South Asia [22]. The prevalence of mental health issues such as depression, anxiety, and stress among the elderly is influenced by socio-economic and health-related factors. Understanding the factors that affect mental health is increasingly important for developing effective policies. This study aimed to observe the mental health disorders condition among elderly people in Bangladesh and to examine the socio-demographic, family, social and health related factor influencing to their mental well-being who were contributing to the financial, family, society, and world.. Understanding how common factors are affecting depression, anxiety, and stress among people aged 60+years in four districts of Bangladesh, and exploring insights into possible solutions to policies about older people.

## Methods and materials

This cross-sectional study was carried out to assess the prevalence of depression, anxiety, and stress levels among the elderly population aged 60 and above in four different districts in Bangladesh. According to $p = 0.5$, $CI = 95\%$(Confidence Interval), $d = 0.05$, the estimated sample size was 384 participants [23,24]. However, a complete sampling frame across four districts was unavailable; therefore, a total of 400 samples were collected using a community-based convenience sampling approach from the Dhaka, Nilphamari, Mymensingh, and Khulna district.s in Bangladesh. The data collectors visited each district and approached eligible individuals aged 60 years and above in households, local markets, parks, and community gathering areas. The inclusion criteria were participants' consent, age 60 years or older, ability to read and write, and provision of informed consent for in-person interviews. On the other hand, the exclusion criteria were participants who were unable to respond to interviews or unwilling to participate in the study. Data collection involved in-person interviews conducted by a trained team using a self-administered questionnaire under the authors' supervision. Data were collected from November 20, 2024, to December 6, 2024. Informed consent was obtained from all participants, and confidentiality and data anonymity were maintained in this study. In this study, the Patient Health Questionnaire (PHQ-9) scale is used to assess depression levels, the Generalized Anxiety Disorder (GAD-7) scale is used to assess anxiety levels, and the Perceived Stress Scale (PSS-10) is used to assess stress symptoms among respondents. Furthermore, the questionnaire included respondents' socio-demographic information, such as age, sex, education level, retirement status, and physical condition. The PHQ-9 and GAD-7 scales were standardized questionnaires that included 9 and 7 questions, respectively, with a 4-point Likert scale for respondents to rate their symptoms. To measure stress symptoms, the PSS-10 scale was used, which consisted of 10 questions on a 5-point Likert scale for the elderly population aged 60 and above. The following classification was used for depression, anxiety, and stress levels according to the PHQ-9, GAD-7, and PSS-10 scale scores:

- Depression level: Minimal: 1–4, Mild: 5–9, Moderate: 10–14, Moderately Severe: 15–19, Severe: 20–27 [25].

- Anxiety level: Minimal: 0–4, Mild: 5–9, Moderate: 10–14, Severe: 15–21 [26].

- Stress level: – Low Stress: 0–13, Moderate Stress: 14–26, High Perceived Stress: 27–40 [27].

Descriptive statistics were used to summarize demographic characteristics and the levels of depression, anxiety, and stress. The association between mental health problems and demographic characteristics was measured using the Pearson chi-square test. To examine the association between explanatory variables and the severity levels of depression, anxiety and stress, the ordinal logistic regression models were fitted. The proportional odds (parallel lines) assumption was assessed using the Test of Parallel Lines. The assumption was satisfied for the depression modeling ($p = 0.568 > 0.05$) and the stress modeling ($p = 0.872 > 0.05$), indicating that ordinal logistic regression was appropriate for these outcomes. However, the assumption was violated for the anxiety modeling ($p = 0.015 < 0.05$), suggesting that the effects of some predictors may vary across anxiety severity categories. To mitigate the issue, a partial proportional odds model was used for anxiety with covariates violating the proportional odds assumption identified using the brant package in R. A Structural equation model (SEM) was employed to confirm the measurement structure of PHQ-9, GAD-7, and PSS-10 scales through a latent variable framework. It's also used to examine how depression, anxiety, and stress interact with the model. SEM allowed us to estimate both direct and indirect pathways and provides additional insight beyond the ordinal regression modeling. Factor analysis was performed using SPSS. Amos software (version 24) was used to fit the SEM model. All descriptive analyses and ordinal regression models were initially fitted using SPSS software (version 25), and due to the violation of the proportional odds assumption for anxiety, additional diagnostic and cross-check analyses were conducted using R software (version 4.5.2).

### Ethics approval and consent to participate

Ethical approval for this study was obtained from the Biosafety, Biosecurity, and Ethical Committee of Jahangirnagar University, Bangladesh, with reference number BBEC, JUM 2024/11 (142), dated November 19, 2024. Additionally, verbal consent was obtained from the participants before the survey began, and the study objectives were explained to them. Participants were sufficiently informed that their data would remain confidential and that no identifiable information would be shared. Additionally, participants may withdraw from the survey at any time. The study is in accordance with national and institutional requirements for ethical research practices and the principles outlined in the Declaration of Helsinki.

## Results

The goal of this study was to determine the covariates and prevalence of mental health conditions of the elderly population aged 60 and above. The distribution of depression, anxiety, and stress levels among people aged 60 years and above is presented in Fig 1.

Fig 1 depicts the percentage distribution of depression levels, anxiety levels, and stress levels among people aged 60 and above. The average depression score was 11.0, with a standard error of 4.480. The figures revealed that 56.5% (26) of the respondents had no depression (minimal depression level). In this study, most of the respondents suffered mild (145) and moderate (139) levels of depression. 17.3% and 5.3% of the respondents, aged 60 years and older, had suffered moderately severe and severe depression, respectively. The average anxiety score was 8.39 with a standard error of 4.331 (according to the GAD-7 scale). The figure shows that 20.0% of the respondents had a minimal level of anxiety. The suffering levels of mild, moderate, and severe anxiety among the respondents were 41.8% (167), 29.0% (116), and 9.3% (37), respectively. Among the 400 respondents aged 60 years and above, the average stress score was 17.47 (PSS-10 scale). Findings revealed that 19.75% (79) of the respondents had low stress. Most of the respondents had a moderate level of stress (75.5%). In this instance, 4.75% (19) of respondents had suffered high perceived stress (Fig 1).

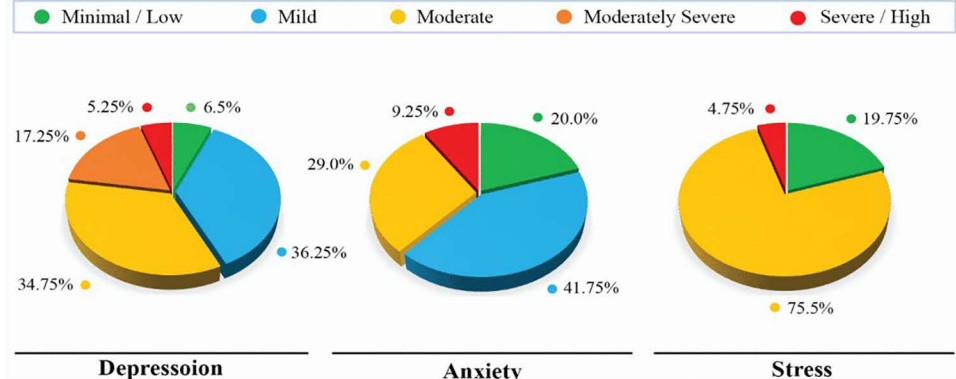

**Fig 1. Prevalence of depression, anxiety, and stress in the elderly population aged 60 and above.**

S1 Table represents the frequency distribution of the respondents by background characteristics. Out of 400 respondents, 363 are male. Approximately 53% of the population is from urban areas, and 47% from rural areas. Most of the respondents had no formal education (44.5%). Twelve percent of the respondents had secondary education qualifications. Among respondents aged 60 and above, 202 (50.5%) lived in joint families. 49.25% of the respondents lived in a single-family home. Among the participants, 129 respondents did not receive any medical or other support from family. Otherwise, 232 and 129 respondents received medical and other support from family, respectively. Among respondents aged 60 and above, 183 reported difficulty accessing transportation, public places, or services. After the age of 60, respondents rated social status as 191 good, 176 fairly, and 33 bad (S1 Table). The Chi-square test was performed to determine the significant characteristics of depression level, anxiety level, and stress level of those aged 60 years and above.

Table 1 presents the prevalence of selected covariate characteristics and their association with the ordinary outcome level of depression. Findings reveal that the depression level had no significant relationship with religion ($p > 0.1$) and retirement status ($p > 0.1$).

Age 60 and older people who lived in rural areas suffered 6.36% minimal, 28.54% mild, 38.1% moderate, 21.18% moderately severe, and 5.82% severe depression. Urban people suffered minimal, mild, moderate, moderately severe, and severe depression 6.63%, 43.15%, 31.75%, 13.73%, and 4.74%, respectively. The significant characteristics of physical condition ($p$-value <0.01) seemed to be associated with depression level. People with good physical health suffered minimal (18.16%), mild (47.76%), moderate (23.84%), moderately severe (9.09%), and severe (1.14%) depression. People with very bad physical health suffer 6.76% mild, 33.3% moderate, 39.95% moderately severe 19.99% severe depression. The proportion of severe depression sufferers increased with physical condition from good to very bad. The study of people aged 60 years and older reveals that 5.85% of people with good family relationships, 11.64% of fair, and 37.38% of those without relationships suffered from severe depression. Among those experiencing social isolation and loneliness, 10.84% suffered from severe levels of depression. Also, 1.64% had severe depression among those who did not experience it (Table 1).

Table 2 presents the prevalence of selected covariate characteristics and their association with the ordinary levels of anxiety and stress. Findings reveal that anxiety level had no significant relationship with sex ($p > 0.1$), religion ($p > 0.1$), education status ($p > 0.1$), and retirement status ($p > 0.1$). More than half of those in very bad physical condition, aged 60 years and older, suffered from severe anxiety (i.e., 53.32%). 22.86%, 40.59%, 29.96%, and 6.59% of respondents with the average physical condition suffer from minimal, mild, moderate, and severe anxiety, respectively. Based on the results, among those who frequently interact with children and grandchildren, 25.64% suffer from minimal, 42.14% mild, 24.89% moderate, and 7.33% severe anxiety. If the importance of opinion in the family decreases, the likelihood of suffering from anxiety increases. Among

**Table 1. Percentage distribution of selected covariates with depression level.**

| Characteristics | Depression | | | | |
| --- | --- | --- | --- | --- | --- |
| | Minimal | Mild | Moderate | Moderately Severe | Severe |
| **Sex\*\*** | | | | | |
| Male | 7.16 | 36.91 | 33.89 | 17.64 | 4.41 |
| Female | – | 29.81 | 43.24 | 13.44 | 13.52 |
| **Religion** | | | | | |
| Islam | 6.38 | 36.17 | 34.83 | 17.03 | 5.59 |
| Hindu | 8.33 | 37.42 | 33.56 | 20.68 | 0 |
| **Residence \*\*** | | | | | |
| Urban | 6.63 | 43.15 | 31.75 | 13.73 | 4.74 |
| Rural | 6.36 | 28.54 | 38.1 | 21.18 | 5.82 |
| **District\*\*\*** | | | | | |
| Dhaka | 13.71 | 37.89 | 28.25 | 15.3 | 4.84 |
| Nilphamari | 6.76 | 37.87 | 24.27 | 22.97 | 8.13 |
| Mymensingh | 1.95 | 27.2 | 45.66 | 21.32 | 3.88 |
| Khulna | 2.02 | 42.44 | 39.42 | 11.07 | 5.04 |
| **Educational status\*\*\*** | | | | | |
| No-education | 4.5 | 26.96 | 38.19 | 21.36 | 8.99 |
| Secondary | 8.13 | 36.06 | 39.57 | 12.75 | 3.49 |
| Higher Secondary | 4.17 | 45.94 | 37.37 | 12.51 | 0 |
| Graduate | 10.86 | 50.14 | 26 | 10.81 | 2.19 |
| Postgraduate | 9.54 | 50.11 | 16.56 | 21.38 | 2.4 |
| **Retirement status** | | | | | |
| Yes | 4.75 | 35.06 | 36.5 | 17.53 | 6.16 |
| No | 8.46 | 37.58 | 32.79 | 16.94 | 4.23 |
| **Physical Condition\*\*\*** | | | | | |
| Good | 18.16 | 47.76 | 23.84 | 9.09 | 1.14 |
| Average | 4.56 | 40.61 | 36.04 | 16.25 | 2.54 |
| Bad | 0.99 | 22.05 | 41.99 | 22.98 | 11.99 |
| Very bad | – | 6.76 | 33.3 | 39.95 | 19.99 |
| **Living status\*\*\*** | | | | | |
| Joint family | 5.44 | 34.16 | 38.11 | 17.83 | 4.46 |
| Single family | 7.62 | 38.58 | 31.48 | 16.75 | 5.59 |
| Old age home | – | 0 | 0 | 0 | 100 |
| **Communication with children and grandchildren\*\*** | | | | | |
| Often | 8.79 | 38.83 | 32.95 | 16.13 | 3.3 |
| Sometimes | 2.04 | 31.64 | 39.83 | 17.31 | 9.19 |
| Less | – | 22.43 | 44.55 | 27.45 | 5.57 |
| None | – | 37.02 | 17.74 | 27.05 | 18.19 |
| **Medical and other support from family\*\*\*** | | | | | |
| Good | 10.78 | 45.24 | 30.19 | 12.07 | 1.72 |
| Fairly | 0.77 | 25.61 | 42.64 | 24 | 6.98 |
| None | – | 17.84 | 35.99 | 25.65 | 20.51 |
| **Financial support to family\*\*\*** | | | | | |
| Yes | 9.02 | 43.14 | 29.42 | 15.28 | 3.14 |
| No | 2.06 | 24.12 | 44.13 | 20.72 | 8.97 |

*(Continued)*

**Table 1.** (Continued)

| Characteristics | Depression | | | | |
|---|---|---|---|---|---|
| | **Minimal** | **Mild** | **Moderate** | **Moderately Severe** | **Severe** |
| **Relationship with family*** | | | | | |
| Good | 8.79 | 42.49 | 34.07 | 13.55 | 1.1 |
| Fairly | 0.96 | 26.18 | 38.86 | 23.31 | 10.68 |
| None | 4.12 | 8.46 | 24.91 | 33.36 | 29.15 |
| **Opinion in the family*** | | | | | |
| Valuable | 8.73 | 41.44 | 33.1 | 15.28 | 1.45 |
| Less | 2.02 | 30.3 | 40.42 | 20.2 | 7.06 |
| None | – | 3.9 | 30.96 | 26.76 | 38.38 |
| **Involvement in social activities** | | | | | |
| Yes | 6.29 | 37.75 | 36.45 | 17.62 | 1.89 |
| No | 6.64 | 35.26 | 33.63 | 17.01 | 7.47 |
| **Communication with neighbors*** | | | | | |
| Often | 8.07 | 41.22 | 33.66 | 13.46 | 3.59 |
| Sometimes | 4.35 | 29.72 | 39.87 | 19.54 | 6.52 |
| Less | 6.92 | 31.06 | 20.65 | 30.99 | 10.38 |
| Very Less | – | 30.21 | 30.34 | 29.44 | 10 |
| **Encountered any discrimination*** | | | | | |
| Yes | 1.01 | 24.56 | 38.72 | 20.42 | 15.3 |
| No | 8.29 | 40.06 | 33.48 | 16.23 | 1.95 |
| **Experienced social isolation and loneliness*** | | | | | |
| Yes | 0 | 29.29 | 36.31 | 23.56 | 10.84 |
| No | 10.7 | 40.75 | 33.74 | 13.17 | 1.64 |
| **Difficulty in transportation, public places or services*** | | | | | |
| Yes | 0.6 | 25.47 | 39.99 | 24.24 | 9.69 |
| No | 10.65 | 43.82 | 31.06 | 12.34 | 2.13 |
| **Changes in social relationships or support*** | | | | | |
| Yes | 2.73 | 28.99 | 33.86 | 26.23 | 8.19 |
| No | 9.68 | 42.38 | 35.5 | 9.67 | 2.77 |
| **Social status *** | | | | | |
| Good | 11.51 | 50.78 | 25.68 | 8.88 | 3.14 |
| Fairly | 2.28 | 26.7 | 44.87 | 21.61 | 4.55 |
| Bad | – | 3.08 | 33.28 | 42.45 | |

**Note:** "*": $p < .05$; "**": $p < 0.01$; "***": $p < 0.001$

those whose opinion is less valuable in the family, 10.1% suffered minimal anxiety, 34.4% were mild, 44.4% were moderate, and 11.1% suffered severe anxiety. 40.01% of people who have difficulty accessing transport, public places, or services due to age suffer from moderate levels of anxiety. On the other hand, respondents who did not face difficulty accessing transportation, public places, or services due to their age suffered minimal, mild, moderate, and severe levels of anxiety, 27.2%, 41.48%, 25.34%, and 5.98%, respectively. The association of stress with different characteristics suggests that stress level was not significantly related to sex, religion, educational status, retirement status, or involvement in social activities ($p > 0.1$).

Findings revealed that 73.87%, 74.1%, 79%, and 79.99% of respondents with good, average, poor, and very poor physical conditions, respectively, report a moderate level of stress. Among the respondents, 77.07% had moderate stress and

**Table 2. Percentage distribution of selected covariates with anxiety and stress level.**

| Characteristics | Anxiety | | | | Stress | | |
|---|---|---|---|---|---|---|---|
| | Minimal | Mild | Moderate | Severe | Low | Moderate | High Perceived |
| **Sex** | | | | | | | |
| Male | 20.67 | 41.31 | 29.75 | 8.27 | 19.28 | 76.6 | 4.13 |
| Female | 13.6 | 45.95 | 21.59 | 18.86 | 24.42 | 64.7 | 10.87 |
| **Religion** | | | | | | | |
| Islam | 20.22 | 40.69 | 29.25 | 9.84 | 19.69 | 75.26 | 5.05 |
| Hindu | 16.63 | 58.3 | 25.07 | 0 | 20.73 | 79.27 | – |
| **Residence** | ** | | | | | | |
| Urban | 25.13 | 41.69 | 25.61 | 7.57 | 21.34 | 74.86 | 3.79 |
| Rural | 14.31 | 41.8 | 32.78 | 11.12 | 17.99 | 76.27 | 5.74 |
| **District** | * | | | | | | |
| Dhaka | 25.83 | 37.07 | 27.43 | 9.68 | 27.43 | 68.53 | 4.04 |
| Nilphamari | 16.2 | 37.88 | 32.42 | 13.49 | 18.88 | 73.01 | 8.11 |
| Mymensingh | 8.78 | 47.5 | 36.94 | 6.79 | 18.49 | 78.6 | 2.92 |
| Khulna | 27.04 | 44.59 | 20.25 | 8.11 | 12.11 | 82.85 | 5.04 |
| **Educational status** | | | | | | | |
| No-education | 16.85 | 41 | 29.78 | 12.37 | 17.98 | 75.84 | 6.18 |
| Secondary | 25.3 | 39.75 | 32.62 | 2.33 | 25.58 | 72.1 | 2.32 |
| Higher Secondary | 20.84 | 45.94 | 24.9 | 8.33 | 14.68 | 83.22 | 2.1 |
| Graduate | 21.71 | 39.15 | 30.47 | 8.67 | 15.25 | 82.56 | 2.18 |
| Postgraduate | 19.01 | 47.62 | 21.5 | 11.87 | 26.23 | 64.2 | 9.58 |
| **Retirement status** | | | | | | | |
| Yes | 17.06 | 42.65 | 28.91 | 11.38 | 18.49 | 75.82 | 5.69 |
| No | 23.29 | 40.74 | 29.1 | 6.87 | 21.16 | 75.14 | 3.7 |
| **Physical Condition** | *** | | | | *** | | |
| Good | 34.04 | 44.34 | 19.35 | 2.27 | 24.99 | 73.87 | 1.15 |
| Average | 22.86 | 40.59 | 29.96 | 6.59 | 21.83 | 74.1 | 4.06 |
| Bad | 5.04 | 42.93 | 35.04 | 16.99 | 13.01 | 79 | 7.99 |
| Very bad | 0 | 13.42 | 33.27 | 53.32 | 6.8 | 79.99 | 13.21 |
| **Living status** | | | | | | | |
| Joint family | 21.78 | 40.1 | 32.68 | 5.44 | 19.8 | 75.25 | 4.95 |
| Single family | 18.27 | 43.65 | 25.38 | 12.7 | 19.8 | 76.14 | 4.06 |
| Old age home | 0 | 0 | 0 | 100 | 0 | 0 | 100 |
| **Communication with children and grandchildren *** | | | | | | | |
| Often | 25.64 | 42.14 | 24.89 | 7.33 | 21.62 | 74.35 | 4.03 |
| Sometimes | 8.16 | 40.86 | 40.8 | 10.18 | 15.33 | 79.57 | 5.1 |
| Less | 5.75 | 44.32 | 33.35 | 16.57 | 11.06 | 77.77 | 11.17 |
| None | 9.44 | 36.38 | 17.9 | 36.27 | 27.4 | 63.4 | 9.19 |
| **Medical and other support from family *** | | | | | | | |
| Good | 28.89 | 41.39 | 22.4 | 7.32 | 22.41 | 73.29 | 4.31 |
| Fairly | 7.76 | 45.74 | 37.98 | 8.53 | 17.84 | 78.28 | 3.88 |
| None | 7.79 | 30.82 | 38.35 | 23.04 | 10.29 | 79.47 | 10.24 |
| **Financial support to family *** | | | | | | | |
| Yes | 25.87 | 42.75 | 23.14 | 8.24 | 21.18 | 74.11 | 4.71 |
| No | 9.66 | 40 | 39.31 | 11.03 | 17.23 | 77.95 | 4.83 |

*(Continued)*

**Table 2.** (Continued)

| Characteristics | Anxiety | | | | Stress | | |
|---|---|---|---|---|---|---|---|
| | Minimal | Mild | Moderate | Severe | Low | Moderate | High Perceived |
| **Relationship with family \*\*\*** | | | | | \*\* | | |
| Good | 27.48 | 40.3 | 26.38 | 5.85 | 22.35 | 74.71 | 2.93 |
| Fairly | 4.89 | 49.45 | 34.01 | 11.64 | 14.58 | 78.63 | 6.79 |
| None | 0 | 25 | 37.62 | 37.38 | 12.55 | 70.69 | 16.76 |
| **Opinion in the family \*\*\*** | | | | | \*\* | | |
| Valuable | 25.46 | 45.81 | 22.91 | 5.81 | 23.27 | 73.46 | 3.27 |
| Less | 10.1 | 34.4 | 44.4 | 11.1 | 13.17 | 81.78 | 5.05 |
| None | 0 | 26.92 | 34.73 | 38.35 | 7.6 | 73.18 | 19.22 |
| **Involvement in social activities** | | | | | | | |
| Yes | 17.01 | 39.61 | 37.73 | 5.66 | 17.58 | 79.9 | 2.52 |
| No | 22 | 43.15 | 23.24 | 11.62 | 21.19 | 72.59 | 6.22 |
| **Communication with neighbors \*\*\*** | | | | | | | |
| Often | 28.26 | 39.46 | 26 | 6.27 | 19.28 | 76.68 | 4.04 |
| Sometimes | 10.9 | 47.07 | 31.18 | 10.86 | 21.01 | 74.64 | 4.35 |
| Less | 6.9 | 38.02 | 34.41 | 20.67 | 17.21 | 68.93 | 13.86 |
| Very Less | 0 | 30.08 | 49.92 | 20 | 20.1 | 79.9 | 0 |
| **Encountered any discrimination \*\*\*** | | | | | | | |
| Yes | 5.13 | 41.84 | 30.6 | 22.42 | 18.4 | 70.36 | 11.24 |
| No | 24.86 | 41.71 | 28.47 | 4.96 | 20.19 | 77.17 | 2.65 |
| **Experienced social isolation and loneliness \*\*\*** | | | | | \* | | |
| Yes | 9.58 | 40.09 | 30.58 | 19.75 | 15.28 | 77.07 | 7.64 |
| No | 26.76 | 42.81 | 27.97 | 2.47 | 22.64 | 74.48 | 2.88 |
| **Difficulty in transportation, public place or services \*\*\*** | | | | | \* | | |
| Yes | 8.49 | 36.34 | 40.01 | 15.16 | 15.74 | 76.99 | 7.27 |
| No | 28.08 | 45.55 | 21.27 | 5.1 | 22.57 | 74.45 | 2.98 |
| **Changes in social relationship or support \*\*\*** | | | | | \* | | |
| Yes | 11.49 | 42.06 | 33.33 | 13.12 | 16.41 | 75.94 | 7.65 |
| No | 27.2 | 41.48 | 25.34 | 5.98 | 22.57 | 75.13 | 2.3 |
| **Social status** | \*\*\* | | | | \* | | |
| Good | 29.87 | 42.41 | 21.44 | 6.28 | 22.49 | 73.33 | 4.19 |
| Fairly | 13.05 | 43.77 | 35.22 | 7.95 | 19.31 | 77.27 | 3.41 |
| Bad | 0 | 27.33 | 39.37 | 33.3 | 6 | 78.84 | 15.17 |

**Note:** "\*": $p < .05$; "\*\*": $p < 0.01$; "\*\*\*": $p < 0.001$

had experienced social isolation and loneliness. As people age, those who had difficulty accessing transportation, public places, or services reported low, moderate, and high perceived levels of stress: 15.74%, 76.99%, and 7.27%, respectively. The Test of Parallel Lines indicated that the proportional odds assumption was met for the depression and stress modeling, but violated for the anxiety modeling. Hence, ordinal logistic regression was used for depression and stress, and a partial proportional odds (multinomial) model was used for anxiety. The effects of key predictors could vary across anxiety severity categories using the partial proportional odds model.

The findings presented in Table 3 indicate that certain characteristics related to the age 60+years, such as physical status, relationship with family as they get older, and social status, are important determinants of mental disorder. The odds

**Table 3. Adjusted odds ratios (AOR) and 95% confidence interval (CI) of adjusted odds ratios (AOR) of ordinal logistic regression for depression, anxiety, and stress status among 60 years and above ages peoples in Bangladesh.**

| Characteristics | Depression | | Anxiety | | Stress | |
|---|---|---|---|---|---|---|
| | AOR | CI [L, U] | AOR | CI [L, U] | AOR | CI [L, U] |
| **Sex** | | | | | | |
| Male | 0.759 | [0.376, 1.530] | | | | |
| Female (Ref.) | – | – | | | | |
| **Residence** | | | | | | |
| Rural | 1.180 | [0.761, 1.830] | 1.445 * | [0.942, 2.217] | | |
| Urban (Ref.) | – | – | – | – | | |
| **District** | | | | | | |
| Dhaka | 0.989 | [0.570, 1.715] | 1.942 ** | [1.127, 3.347] | 0.462** | [0.238, 0.894] |
| Nilphamari | 1.254 | [0.663, 2.373] | 1.972 ** | [1.048, 3.713] | 0.735 | [0.339, 1.593] |
| Mymensingh | 0.838 | [0.461, 1.525] | 0.973 | [0.527, 1.797] | 0.511 * | [0.244, 1.070] |
| Khulna (Ref.) | – | – | – | – | – | – |
| **Education Level** | | | | | | |
| No-education | 0.941 | [0.457, 1.938] | | | | |
| Secondary | 0.572 | [0.265, 1.235] | | | | |
| Higher Secondary | 0.808 | [0.350, 1.866] | | | | |
| Graduate | 0.567 | [0.243, 1.325] | | | | |
| Postgraduate (Ref.) | – | – | | | | |
| **Physical status** | | | | | | |
| Good | 0.206 *** | [0.066, 0.647] | 0.241 ** | [0.076, 0.763] | 0.492 | [0.116, 2.085] |
| Average | 0.390 * | [0.131, 1.157] | 0.367 * | [0.122, 1.100] | 0.603 | [0.150, 2.426] |
| Bad | 0.796 | [0.262, 2.421] | 0.625 | [0.203, 1.919] | 0.950 | [0.231, 3.900] |
| Very bad (Ref.) | – | – | – | – | – | – |
| **Financial support to family** | | | | | | |
| No | 1.763 ** | [1.136, 2.735] | | | | |
| Yes (Ref.) | – | – | – | – | | |
| **Communication with children and grandchildren** | | | | | | |
| Often | 4.332 ** | [1.127, 16.650] | 1.662 | [0.447, 6.177] | | |
| Sometimes | 3.049 | [0.763, 12.184] | 1.619 | [0.420, 6.238] | | |
| Less | 2.280 | [0.479, 10.861] | 1.398 | [0.304, 6.420] | | |
| None (Ref.) | – | – | – | – | | |
| **Medical and other support from family** | | | | | | |
| Good | 0.513 | [0.22, 1.198] | 0.939 | [0.401, 2.197] | | |
| Fairly | 1.056 | [0.465, 2.401] | 1.150 | [0.503, 2.628] | | |
| None (Ref.) | – | – | – | – | | |
| **Relationship with family as get older** | | | | | | |
| Good | 0.782 | [0.236, 2.589] | 0.437 | [0.132, 1.446] | 1.540 | [0.380, 6.240] |
| Fairly | 1.442 | [0.466, 4.467] | 0.643 | [0.208, 1.994] | 2.090 | [0.535, 8.163] |
| None (Ref.) | – | – | – | – | – | – |
| **Opinion in the family** | | | | | | |
| Valuable | 0.207 *** | [0.069, 0.624] | 0.371 * | [0.125, 1.098] | 0.341 | [0.083, 1.394] |
| Less | 0.190 *** | [0.064, 0.569] | 0.736 | [0.252, 2.155] | 0.620 | [0.156, 2.471] |
| None (Ref.) | – | – | – | – | – | – |

*(Continued)*

**Table 3.** (Continued)

| Characteristics | Depression | | Anxiety | | Stress | |
|---|---|---|---|---|---|---|
| | AOR | CI [L, U] | AOR | CI [L, U] | AOR | CI [L, U] |
| **Involvement in social activities** | | | | | | |
| No | | | 0.472 *** | [0.306, 0.727] | | |
| Yes (Ref.) | | | – | – | | |
| **Communication with neighbors** | | | | | | |
| Often | 3.397 * | [0.907, 12.726] | 0.590 | [0.159, 2.189] | | |
| Sometimes | 2.441 | [0.658, 9.058] | 0.630 | [0.17, 2.338] | | |
| Less | 3.016 | [0.712, 12.775] | 1.169 | [0.276, 4.958] | | |
| Very Less (Ref.) | – | – | – | – | | |
| **Encountered discrimination or prejudice** | | | | | | |
| No | 1.907 ** | [1.058, 3.437] | | | 1.564 | [0.77, 3.175] |
| Yes (Ref.) | – | – | – | – | – | – |
| **Experienced social isolation and loneliness** | | | | | | |
| No | 0.429 *** | [0.265, 0.693] | | | 0.724 | [0.405, 1.297] |
| Yes (Ref.) | – | – | – | – | – | – |
| **Difficulty accessing transportation, public place or services** | | | | | | |
| No | 0.513 *** | [0.322, 0.816] | 0.586 ** | [0.37, 0.929] | 0.829 | [0.473, 1.453] |
| Yes (Ref.) | – | – | – | – | – | – |
| **Changes in social relationship or support** | | | | | | |
| No | 0.621 ** | [0.393, 0.982] | 1.053 | [0.672, 1.651] | 0.706 | [0.408, 1.221] |
| Yes (Ref.) | – | – | – | – | – | – |
| **Social status** | | | | | | |
| Good | 0.163 *** | [0.067, 0.398] | 0.320 ** | [0.133, 0.771] | 0.413 | [0.134, 1.269] |
| Fairly | 0.400 ** | [0.176, 0.912] | 0.414 ** | [0.18, 0.95] | 0.362 * | [0.123, 1.065] |
| Bad (ref.) | – | – | – | – | – | – |

**Note:** Ref.: Reference category, "*": $p < 0.05$; "**": $p < 0.01$; "***": $p < 0.001$

of higher depression increase with physical health condition, with good health condition being 0.206 times (OR: 0.206, 95% CI: 0.066, 0.647) lower and 0.390 times (OR: 0.390, 95% CI: 0.131, 1.157) lower for average physical health condition, likely to be depressed compared to the very bad physical health condition. Furthermore, people aged 60+ who were not giving financial support to the family had a higher probability of depression, with giving financial support to the family associated with a 1.76 times (OR: 1.763, 95% CI: 1.136, 2.735) higher likelihood. Regarding opinion in the family aged 60+, the odds of higher depression for valuable opinion were 0.207 times (OR: 0.207, 95% CI: 0.069, 0.624) lower depression level compared with no opinion in the family. Similarly, the odds of higher depression for less valuable opinions in the family were 0.190 times (OR: 0.190, 95% CI: [0.064, 0.569]) lower depression level compared with no opinion in the family. People aged 60+ who had not experienced social isolation and loneliness had 0.429 times (OR: 0.429, 95% CI: 0.265, 0.693) lower depression compared with experienced social isolation and loneliness. However, the odds of higher depression for people aged 60+ years who had not faced difficulty accessing transportation, public places, or services were 0.513 times (OR: 0.513, 95% CI: 0.322, 0.816) lower than those who faced difficulty accessing transportation, public places, or services. At the age of 60+ years, changes in social relationships and support had a higher probability of depression, compared with those who had none, 0.621 times (OR: 0.621, 95% CI: 0.393, 0.982) lower depression. However, good social status was 0.163 times (OR: 0.163, 95% CI: 0.067, 0.398) lower, and fairly social status was 0.400 times (OR: 0.400, 95% CI: 0.176,

0.912) lower depression compared with bad social status. The odds of higher anxiety for those who were not involved in social activities were 0.472 times (OR: 0.472, 95% CI: 0.306, 0.727) lower than those who were involved in social activities. However, those aged 60+who had not experienced social isolation and loneliness had 0.499 times (OR: 0.499, 95% CI: 0.311, 0.801) lower anxiety compared with those who experienced social isolation and loneliness. Difficulty accessing transportation, public places, or services had 0.414 times (OR: 0.586, 95% CI: 0.37, 0.929) higher anxiety compared with no experience.

Table 4 shows that people aged 60 and above who did not provide financial support to their family had higher had higher odds of reporting mild (OR = 2.27; 95% CI: 1.16–4.43), moderate (OR = 4.11; 95% CI: 2.06–8.21), and severe anxiety (OR = 2.63; 95% CI: 1.04–6.70) compared with those who did, indicating particularly strong effects at the moderate level (p<0.01). Older adults who not experiencing discrimination or prejudice was associated with lower of odds of mild (OR = 0.30; 95% CI: 0.11–0.83), moderate (OR = 0.32; 95% CI: 0.11–0.93), and severe anxiety (OR = 0.14; 95% CI: 0.04–0.47), while not experiencing social isolation and loneliness significantly reduced the odds of severe anxiety (OR = 0.09; 95% CI: 0.03–0.28), with weaker and non-significant effects for mild and moderate anxiety.

A measurement model was developed comprising four latent constructs to explore the primary factors affecting the mental health of individuals aged 60 and above, and presented in Fig 2. Confirmatory factor analysis (CFA) indicated that the latent constructs depression, anxiety, stress, and socio-demographic factors are represented by observed variables DeS, AS, SS, and SD, respectively, with factor loadings indicating their relationships. Items with factor loadings below 0.5 were excluded to ensure construct reliability, validity, and model fit [28]. Construct reliability and validity were assessed with Composite Reliability (CR) values ranging from 0.692 to 0.775, surpassing the 0.70 threshold [29]. Cronbach's coefficient alpha scores above 0.70 indicate a model is reliable and valid [30]. Tests of reliability and validity are presented in S2 Table and confirm that all constructs and their indicators are reliable and valid for further analysis.

Sixteen indicators of the conceptual framework model were analyzed for hypothesis testing (Fig 2). Before finalizing the SEM model, a measurement model was employed to validate the reliability and validity of these indicators (S2 Table).

The normalized $\chi^2$ ($\chi^2$/df) is utilized to assess the lack of fit, indicating a well-fitted model. NFI values range from 0 to 1, and values above 0.90 suggest a good fit. Similarly, CFI values range from 0.0 to 1.0, with values greater than 0.90 indicating a good fit. AGFI adjusts GFI based on degrees of freedom, with values above 0.90 generally indicating well-fitting models. In a well-fitting model, RMSEA serves as a secondary fit index, with the lower limit approaching 0 and the upper limit less than 0.08. In this study, these fit indices collectively suggest that the model fits well (Table 5). Confirmatory factor analysis (CFA) was performed to validate the structure of the PHQ-9, GAD-7, and PSS-10 scales. Structural equation modeling (SEM) was also employed to confirm the factor structure of these instruments and to

**Table 4. Odds ratios (OR) and 95% confidence interval of odds ratios of multinomial regression coefficient for anxiety among individuals aged 60 years and above in Bangladesh.**

| Characteristics | Mild | | Moderate | | Severe | |
|---|---|---|---|---|---|---|
| | OR | CI [L, U] | OR | CI [L, U] | OR | CI [L, U] |
| Financial support to the family | | | | | | |
| No | 2.270 * | [1.162, 4.431] | 4.108 ** | [2.056, 8.207] | 2.634 * | [1.036, 6.695] |
| Yes (Ref.) | | | | | | |
| Encountered discrimination or prejudice | | | | | | |
| No | 0.295 * | [0.105, 0.827] | 0.317 * | [0.108, 0.925] | 0.142 ** | [0.043, 0.471] |
| Yes (Ref.) | | | | | | |
| Experienced social isolation and loneliness | | | | | | |
| No | 0.563 | [0.281, 1.125] | 0.489 | [0.234, 1.023] | 0.093 *** | [0.030, 0.284] |
| Yes (Ref.) | | | | | | |

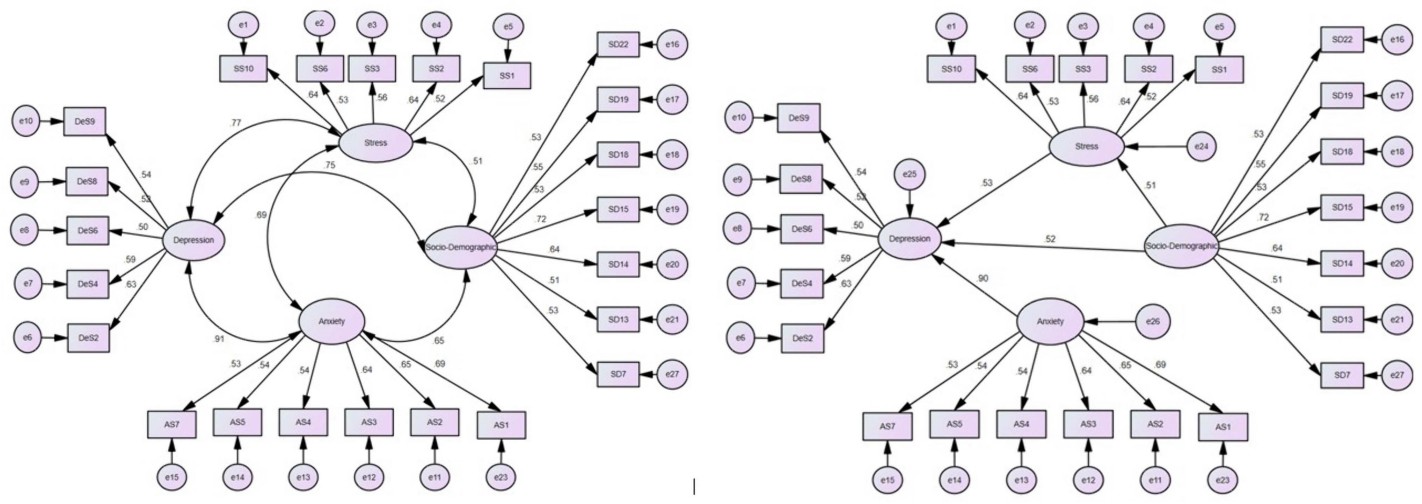

**Fig 2. Measurement model (left panel), Structural equation model (right panel).**

examine the relationships among scale items and their underlying factors. Path coefficients from the SEM model are depicted in Fig 2. The findings illustrate the interconnectedness among the three components measuring mental health (Table 6).

The path coefficients of the structural equation model (SEM) indicated that the first four null hypotheses were rejected, whereas the fifth was accepted. Hypothesis $H_{01}$ : (Socio-demographic factors are positively associated with the stress of people aged 60 and above) is accepted ($\beta$ =0.821, $p$ <0.001). Hypothesis $H_{02}$ : (Socio-demographic factors are positively associated with depression in people aged 60 and above) have an acceptable strength ($\beta$ =0.816, $p$ <0.001). The third hypothesis $H_{03}$ : (Stress is positively related to the depression of people aged 60 and above) was accepted ($\beta$ =0.565, $p$ <0.001). Hypothesis $H_{04}$ : (Depression is positively related to the anxiety of people aged 60 and above) had acceptable strength. The fifth hypothesis $H_{05}$ : (Stress is positively related to the anxiety of people aged 60 and above) was rejected, and the null hypothesis was accepted (Table 5).

**Table 5. Model fit index of confirmatory factor analysis.**

| Goodness-of-fit indices | Model Value | Recommended Value |
|---|---|---|
| $\chi^2/df$ | 2.063 | ≤ 3 |
| NFI | 0.914 | ≥ 0.90 |
| IFI | 0.943 | ≥ 0.90 |
| TLI | 0.927 | ≥ 0.90 |
| CFI | 0.937 | ≥ 0.90 |
| PNFI | 0.628 | ≥ 0.50 |
| PCFI | 0.682 | ≥ 0.50 |
| RMSEA | 0.067 | < 0.08 |
| AIC | 771.241 | Lower values indicate a better fit |

**Note:** NFI: Normed Fit Index, IFI: Incremental Fit Index, TLI: Tucker-Lewis Index, CFI: Comparative Fit Index, PNFI: Parsimonious Normed Fit Index, PCFI: Parsimonious Comparative Fit Index, RMSEA: Root Mean Square Error of Approximation, AIC: Akaike Information Criterion.

**Table 6. Hypothesis test result based on structural equation model (SEM).**

| Hypothesis | Hypothesis Path | Estimate | S. E | Decision |
|---|---|---|---|---|
| $H_1$ | Socio-Demographic ---> Stress | 0.821*** | 0.159 | Accepted |
| $H_2$ | Socio-Demographic ---> Depression | 0.816*** | 0.152 | Accepted |
| $H_3$ | Stress ---> Depression | 0.565*** | 0.099 | Accepted |
| $H_4$ | Depression ---> Anxiety | 0.941*** | 0.137 | Accepted |
| $H_5$ | Stress ---> Anxiety | 0.020 | 0.124 | Rejected |

**Note:** $p < 0.05$: *, $p < 0.01$: **, $p < 0.001$: ***

## Discussion

The findings revealed that the respondents had suffered from depression of 93.5% (36.25% mild, 34.75% moderate, 17.25% moderately severe, and 5.25% severe), anxiety of 80% (41.75% mild, 29.0% moderate, and 9.25% severe), and stress of 80.25% (75.5% moderate and 4.75% severe). The respondents had an average score of depression of $11.0 \pm 4.78$ (PHQ-9 scale), anxiety of $8.39 \pm 4.331$ (GAD-7 scale), and stress of $17.47 \pm 5.17$ (PSS-10 scale). The findings of severe and moderate levels of depression were lower than those of other studies [13,31–35], however, they are higher than in some previous studies [7,9]. The findings of severe and moderate levels of anxiety were also higher than those of previous studies [13,19,35–37] and lower than those of other studies [7,15]. The moderate-high stress level rates are lower than those reported in the study of women [26,27] and higher than those reported in the study [13].

It is observed that several socio-demographic characteristics were significantly associated with depression, anxiety, and stress levels. The findings revealed that the significant prevalence of severe depression, anxiety, and stress among the respondents with physical conditions was 51.1%, 45.9%, and 42.1%, respectively. However, previous studies also revealed that physical condition had a significant association with depression, anxiety, and stress [14,18,38]. The study in Khoy city had a significant prevalence of 44.3% for depression and 31.4% for anxiety among those with poor health conditions. There was no significant association between stress and health conditions [13]. This study showed that living status had a significant association with depression, stress, and anxiety. The prevalence of depression, stress, and anxiety for respondents living alone in this study was higher than in the previous study, which revealed the prevalence to be 11.4%, 10.0%, and 1.4%, respectively, by [13]. Conversely, respondents living with family had a significant association with depression and anxiety in this study, which was also found to be significant in previous studies [39]. The education level was found to be a significant characteristic influencing depression symptoms in this study. Higher education was associated with lower depression, whereas lower education was associated with higher depression. Anxiety and stress were not significantly influenced by education level. However, previous studies showed a significant association with anxiety and stress, as well as depression [7,13]. A previous study [40] showed significance with depression, and the study [11] showed significance with stress.

The findings show that social activity has a significant relationship with mental health conditions. For example, in the elderly, social relationships or support had a significant association with depression, anxiety, and stress. Similarly, previous studies have shown that social support has a significant association with mental health conditions [11,14,32,38]. People who were socially isolated or alone suffered from depression and anxiety significantly. In this study, elderly people who also faced social prejudice and discrimination suffered from depression, anxiety, and stress significantly. Researchers found that social isolation or being alone influenced depression [41]. Findings depict that 61.9% of people with severe depression and 43.2% with severe anxiety were significantly affected at the age of 60 and above, particularly those who did not receive financial support from their families. However, stress levels did not show a significant association with financial support. The previous study [13] reported that the levels of depression, anxiety, and stress were 25.2%, 20.4%, and 8.7%, respectively, depending on financial support from the family. However, there was no significant association between stress levels

and family financial support. In some situations, our findings differ from those of other studies, and one possible reason is sampling variability.

## Limitations of the study

The results may be subject to biases from self-reported responses and to sampling variability. The causal inference is not possible because this is a cross-sectional study. The study sample contained a disproportionately high number of male respondents (363 of 400), which may reflect gendered patterns in availability during data collection and social norms that restrict elderly women's participation in household-based surveys. This imbalance should be interpreted with caution and may limit the generalizability of gender-based findings. Some covariates showed very wide confidence intervals for their odds ratios, reflecting low statistical precision. These imprecise estimates, such as those for frequent communication with children and grandchildren, should therefore be interpreted with caution and regarded as exploratory rather than definitive evidence of association.

## Conclusion

Older adults in Bangladesh (aged 60 years and above) experience substantial levels of depression, anxiety, and stress in later life. At the age of 60 years or more, those in poor health conditions, living alone, without social support, or experiencing social isolation had a higher risk of depression, anxiety, and stress. Risk factors such as health conditions, communication with children, medical support from family, relationship with the family as they get older, social discrimination and prejudice, and perceived social status contributed to depression, anxiety, and stress. The CFA indicates that the underlying factors are interrelated among PHQ-9, GAD-7, and PSS-10. The harmful factors causing mental health problems in older people need to be identified and addressed through appropriate interventions. Families and society must create a comfortable and accessible environment for them. Policymakers should develop policies to support older people. They have given their best to their family, society, country, and the world. We should take care of them in the final parts of our lives.

## Supporting information

**S1 Table. Frequency n (%) of individuals aged 60 years and older by socioeconomic characteristics in Bangladesh.**
(DOCX)

**S2 Table. Test of Reliability and Validity.**
(DOCX)

**S1 File. Data file.**
(CSV)

## Acknowledgments

All authors are grateful to the participants for their time and contributions to this study and to those who helped in data collection. The authors are also thankful to the Biosafety, Biosecurity, and Ethical Approval Committee at Jahangirnagar University for providing ethical oversight and clearance for this study. We are thankful to the Academic Editor and the anonymous reviewers for their valuable comments and feedback, which helped enhance the quality of the manuscript.

## Author contributions

**Conceptualization:** Sahabul Haque, Lamia Jannat, Imran Hossain Sumon, Md. Moyazzem Hossain.

**Data curation:** Sahabul Haque, Suchana Akter, Lamia Jannat, Zidan Ahmed, Mohammad Arifur Rahman.

**Formal analysis:** Sahabul Haque, Suchana Akter, Mohammad Arifur Rahman, Imran Hossain Sumon.

**Methodology:** Sahabul Haque, Imran Hossain Sumon, Md. Moyazzem Hossain.

**Supervision:** Md. Moyazzem Hossain.

**Validation:** Md. Moyazzem Hossain.

**Visualization:** Sahabul Haque, Mohammad Arifur Rahman, Imran Hossain Sumon, Md. Mahfuzur Rahman, Md. Salah Uddin.

**Writing – original draft:** Sahabul Haque, Suchana Akter, Lamia Jannat, Zidan Ahmed, Mohammad Arifur Rahman, Imran Hossain Sumon, Md. Mahfuzur Rahman, Md. Salah Uddin.

**Writing – review & editing:** Md. Moyazzem Hossain.

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
