## [Decision Letter · Decision Letter 0]

27 Nov 2025

Dear Dr. Moyazzem,

Thank you for submitting your manuscript to PLOS ONE. After careful consideration, we feel that it has merit but does not fully meet PLOS ONE’s publication criteria as it currently stands. Therefore, we invite you to submit a revised version of the manuscript that addresses the points raised during the review process.

We look forward to receiving your revised manuscript.

Kind regards,

Md Saiful Islam, BPH, MPH

Academic Editor

PLOS ONE

2. In the online submission form you indicate that your data is not available for proprietary reasons and have provided a contact point for accessing this data. Please note that your current contact point is a co-author on this manuscript. According to our Data Policy, the contact point must not be an author on the manuscript and must be an institutional contact, ideally not an individual. Please revise your data statement to a non-author institutional point of contact, such as a data access or ethics committee, and send this to us via return email. Please also include contact information for the third party organization, and please include the full citation of where the data can be found.

Additional Editor Comments:

It’s not necessary to cite a paper unless you feel it is required, even if a reviewer suggests adding one.

Reviewers' comments:

Reviewer's Responses to Questions

**Comments to the Author**

1. Is the manuscript technically sound, and do the data support the conclusions?

Reviewer #1: Yes

Reviewer #2: No

Reviewer #3: Partly

2. Has the statistical analysis been performed appropriately and rigorously?

Reviewer #1: Yes

Reviewer #2: No

Reviewer #3: Yes

3. Have the authors made all data underlying the findings in their manuscript fully available?

Reviewer #1: Yes

Reviewer #2: No

Reviewer #3: Yes

4. Is the manuscript presented in an intelligible fashion and written in standard English?

Reviewer #1: Yes

Reviewer #2: No

Reviewer #3: No

Reviewer #1: The manuscript is generally well written. The authors would benefit by incorporating the following articles in the introduction and discussion sections-

https://doi.org/10.1007/978-981-99-9153-2_11

https://doi.org/10.9790/0837-2108025873

In the discussion, the authors acknowledge their results differ from some previous studies but don't deeply explore why these differences exist beyond general contextual factors. The manuscript could have benefited from more discussion about the cultural context of mental health in Bangladesh. Most importantly, study identifies important risk factors like social isolation and poor health status, it doesn't adequately address the potential confounding variables or explore the complex interplay between multiple risk factors.

Reviewer #2: Thank you for the opportunity to review this manuscript, which examines the important topic of mental health among the elderly in Bangladesh. The use of validated tools (PHQ-9, GAD-7, PSS-10) and the attempt to use advanced statistical methods (SEM) are noted strengths.

However, I must express serious concerns regarding the rigor, preparation, and reporting of this study. I encountered several major errors in the text that appear to be careless copy-paste remnants from other documents. These errors severely undermine my confidence in the carefulness of the entire work. Furthermore, the methodology lacks critical details necessary to verify the claim of "random" sampling, and the statistical reporting needs significant improvement.

I cannot recommend this manuscript for publication in its current state. It requires a major overhaul to address these fundamental issues.

Specific Major Concerns:

1. I was alarmed to find text in the Results section that has absolutely nothing to do with this study. On page 4, under the 'Results' heading, the very first paragraph ends with: "The prevalence of stunted, underweight, and wasted children and their composition are presented in Figure 1." This is a study about the elderly, not malnourished children. The presence of such a glaring error suggests that the manuscript was not thoroughly proofread before submission. Furthermore, in the Discussion (Page 18), there is a placeholder left in the text: "...revealed the prevalence to be 11.4%, 10.0%, and 1.4%, respectively, by [NO_PRINTED_FORM] Conversely..." These errors are unprofessional and must be corrected. They also lead me to question if other parts of the data or text were inadvertently carried over from different projects.

2. The authors state: "total of 400 elderly population aged 60 and above were selected using stratified random samples from the Dhaka, Nilphamari, Mymensingh, and Khulna districts." This is a very strong claim that is currently unsupported by the text. To conduct true stratified random sampling, you would need a complete sampling frame (a list of every single elderly person in these four districts) from which to randomly draw participants. Did such a list exist? If you did not use a master list, this is likely not random sampling, but rather convenience or purposive sampling (perhaps with quotas for each district). You must honestly describe exactly how a participant was recruited. Did you go door-to-door? Did you visit clinics? "Random" has a specific statistical meaning; if standard randomization procedures weren't followed, you must change this terminology and list it as a major limitation, as it affects the generalizability of your prevalence findings.

3. The sample size calculation needs clarification. You state: "According to p=0.05, CI=95% (Confidence Interval), d=0.05, and a power of 80%, the estimated sample size was 383 participants (16)." Usually, in prevalence studies, 'p' stands for the expected prevalence. Did you really expect only a 5% prevalence (p=0.05) of mental health issues? If the expected prevalence is unknown, it is standard practice to use p=0.5 (50%) to yield the maximum sample size for a given precision. If you used p=0.05 to get N=383, your study might be underpowered if the actual prevalence is much higher (which your results show it is). Please clarify this calculation.

4(a). You have used ordinal logistic regression, which is appropriate for your ranked outcome data. However, this model assumes the "proportional odds" assumption (i.e., the relationship between predictors and the outcome is the same across all cut-points of the severity scale). You must test this assumption and report whether it was met. If it was violated, a different model (like multinomial logistic regression) might be needed.

4(b). Some of your Confidence Intervals in Table 3 are very wide (e.g., Communication with children 'Often': OR 4.332, CI [1.127, 16.650]). This indicates low precision, likely due to small cell counts in those specific categories. You should acknowledge this lack of precision in your interpretation.

5. While SEM is a powerful tool, it is not entirely clear what unique value it adds here beyond the regression analysis, other than confirming that Depression, Anxiety, and Stress are correlated (which is already well-known). The hypotheses tested (H1-H5) are very basic. Please provide a stronger rationale in the introduction or methods for why SEM was necessary to answer your specific research questions.

Minor (but important) Concerns:

1. The abstract mentions "unknown risk factors" in the background. Many risk factors for elderly depression are actually quite well known globally. It would be better to frame this as a need to understand these factors specifically within the Bangladeshi context.

2. The manuscript needs careful proofreading. Many sentences are grammatically incorrect or poorly phrased, making them difficult to understand.

(a) Example (Introduction): "This study aimed to find the people who were contributing to the financial, family, society, and world, and how they were spending their lives with mental health." This sentence is convoluted and its meaning is unclear.

(b) Example (Conclusion): "Older people (aged 60 years and above) were in depression, anxiety, and stress in the end stage of life." The phrasing "were in depression" is awkward, and "end stage of life" is a very strong and likely inaccurate generalization.

(c) Example (Conclusion): "The harmful factors causing mental health problems in older people need to be diagnosed." Factors are identified or addressed, not diagnosed.

(d) Repetitive Text: The manuscript repeatedly uses the same awkward phrasing. For example, in Table 2's description, the same sentence about non-significant findings is repeated three times in a single paragraph.

Reviewer #3: The article entitled “Prevalence and Determinants of Depression, Anxiety, and Stress Among the Elderly Population in Bangladesh: A Cross-sectional Study” employs a cross-sectional design that is methodologically consistent with the aims of the investigation. However, the gender distribution warrants attention: 363 of the 400 respondents are male, resulting in a markedly homogeneous sample. This imbalance introduces a potential sampling bias that limits the representativeness of the results and should be explicitly recognized as a methodological weakness.

The ethical procedures described appear adequate; nonetheless, it should be noted that in several countries verbal consent is not considered sufficient, and the authors could provide more detail regarding how this process was conducted.

The statistical analyses were performed appropriately, ensuring analytical rigor and supporting the validity of the reported findings. The results are coherent and provide empirical support for the study’s conclusions.

With regard to the references, 27 of the 31 citations are more than five years old, which suggests that the manuscript would benefit from the inclusion of more recent literature to strengthen its theoretical and empirical foundation.

Although the manuscript is understandable, the standard of English requires substantial revision. Several sections contain unclear or disconnected statements, such as:

• In the abstract, the results section includes the sentence “Significant risk factors include medical support from family, relationships with family as

• age progresses,” which is confusing. A more precise formulation would be: “Significant risk factors include poor medical support from family and inadequate family relationships as age progresses.”

• In the final paragraph of the introduction, the sentence “This study aimed to find the people who were contributing to the financial, family, society, and world, and how they were spending their lives with mental health” is not aligned with the surrounding text and lacks clarity.

• In the first paragraph of the Results section, the statement “The prevalence of stunted, underweight, and wasted children and their composition are presented in Figure 1” is inconsistent with the study’s focus on the elderly population and appears to be an oversight.

In summary, despite the limitations observed, particularly regarding sampling composition and language clarity, the study contributes relevant findings to its field and addresses an important public health issue.

**Do you want your identity to be public for this peer review?** For information about this choice, including consent withdrawal, please see our Privacy Policy

Reviewer #1: No

Reviewer #2: No

Reviewer #3: No

---

## [Author Response · Author response to Decision Letter 1]

21 Jan 2026

We would like to sincerely thank the anonymous reviewers and the Academic Editor for their valuable comments. We have considered all comments and then thoroughly revised and formatted the manuscript. A detailed response to each comment is provided below.

Author's Response to the Editor Comments:

Response Note

Thanks. We appreciate your feedback. As per comments, a careful revision has been conducted, and all required files are uploaded to the journal submission system. The revised texts are highlighted in “red” color.

Response Note

Thank you very much, we revised the manuscript following the PLOS ONE style.

The revised texts are in “red”. Page: 18-23

Thanks. We revised the data available statement. We also submit the data file as a supplementary file. Page: 23

Thank you very much for your insightful comments and feedback. We added the ethics statement in the Methodology section. Page: 5

Author's Response to the Reviewer 1 Comments:

Thank you very much for your insightful comments and feedback. We believe that it helps to enhance the quality of the manuscript. We revised the Introduction section and cited the suggested papers. The revised texts are in “red”. Page: 2

Thanks. The main reason for the different results may be sampling variability. We added it to the Discussion section and the Limitations section. Page: 22

Author's Response to the Reviewer 2 Comments:

Thank you very much for your comments and feedback. We believe that it helps to improve the quality of the manuscript.

We revised the manuscript accordingly. Page: 4

Thanks. We revised the Results section. Page:6, 22

Thanks. We revised the Methods section. Page:4.

Thank you very much. We revised the Methods section. Page:4.

Thanks. We revised the Methods section. page: 5.

Thanks. We agreed with you. We mentioned it in the limitation section. This may be happed due to small sample of that category. page:23

Thanks. We revised the Methods section. page: 5

Thank you very much. We revised the Abstract of the manuscript. page:1

Thank you. We checked the entire manuscript and corrected grammatical issues and typos. page:1-23

Thanks. We revised the Conclusion section. page:23.

Thanks. We revised the Conclusion section. page:23.

Thanks. We revised the manuscript accordingly. page:11,14.

Author's Response to the Reviewer 3 Comments:

Thank you very much for carefully checking the manuscript and providing comments and feedback. We believe that this helps to improve the quality of the manuscript. The limitatioare mentioned in “Blue” page:23.

Thanks. We revised the ethical declaration. Page: 4

Thanks for your positive comments. We appreciate it,

Thanks. We added some recent citations.

Thank you very much for your careful checking. We revised the Abstract as per your suggestion. Page: 2

Thanks. We revised the Introduction section. page:3

Thanks. We revised the results section. Page:6

In conclusion, the revised version of the manuscript has been produced as per the review outcomes. So, we hope that you will be happy to see this greatly improved version. Once again, we would like to thank you all for your dedication, professional services, and cooperation.

---

## [Decision Letter · Decision Letter 1]

6 Mar 2026

Prevalence and Determinants of Depression, Anxiety, and Stress Among the Elderly Population in Bangladesh: A Cross-sectional Study

PONE-D-25-55520R1

Dear Dr. Hossain,

We’re pleased to inform you that your manuscript has been judged scientifically suitable for publication and will be formally accepted for publication once it meets all outstanding technical requirements.

Kind regards,

Md Saiful Islam, BPH, MPH

Academic Editor

PLOS One

Reviewers' comments:

Reviewer's Responses to Questions

**Comments to the Author**

Reviewer #2: All comments have been addressed

2. Is the manuscript technically sound, and do the data support the conclusions?

Reviewer #2: Yes

3. Has the statistical analysis been performed appropriately and rigorously?

Reviewer #2: Yes

4. Have the authors made all data underlying the findings in their manuscript fully available?

Reviewer #2: Yes

5. Is the manuscript presented in an intelligible fashion and written in standard English?

Reviewer #2: Yes

Reviewer #2: The authors have successfully addressed the concerns raised during the review process, and the manuscript is now suitable for publication. The manuscript now presents a cohesive and scientifically sound narrative that offers meaningful insights into the mental health of the elderly population in Bangladesh.

**Do you want your identity to be public for this peer review?** For information about this choice, including consent withdrawal, please see our Privacy Policy

Reviewer #2: No

---

## [Editor Report · Acceptance letter]

PONE-D-25-55520R1

PLOS One

Dear Dr. Hossain,

I'm pleased to inform you that your manuscript has been deemed suitable for publication in PLOS One. Congratulations! Your manuscript is now being handed over to our production team.

Kind regards,

on behalf of

Mr. Md Saiful Islam

Academic Editor

PLOS One